# UPR: An Upstream Signal to EMT Induction in Cancer

**DOI:** 10.3390/jcm8050624

**Published:** 2019-05-08

**Authors:** Patricia G. Santamaría, María J. Mazón, Pilar Eraso, Francisco Portillo

**Affiliations:** 1Departamento de Bioquímica, Universidad Autónoma de Madrid, 28029 Madrid, Spain; pgsantamaria@iib.uam.es (P.G.S.); peraso@iib.uam.es (P.E.); 2Instituto de Investigaciones Biomédicas “Alberto Sols” CSIC-UAM, 28029 Madrid, Spain; mjmazon@iib.uam.es; 3IdiPAZ, 28029 Madrid, Spain; 4CIBERONC, 28029 Madrid, Spain

**Keywords:** epithelial-mesenchymal transition, endoplasmic reticulum stress, metastasis, plasticity, unfolded protein response

## Abstract

The endoplasmic reticulum (ER) is the organelle where newly synthesized proteins enter the secretory pathway. Different physiological and pathological conditions may perturb the secretory capacity of cells and lead to the accumulation of misfolded and unfolded proteins. To relieve the produced stress, cells evoke an adaptive signalling network, the unfolded protein response (UPR), aimed at recovering protein homeostasis. Tumour cells must confront intrinsic and extrinsic pressures during cancer progression that produce a proteostasis imbalance and ER stress. To overcome this situation, tumour cells activate the UPR as a pro-survival mechanism. UPR activation has been documented in most types of human tumours and accumulating evidence supports a crucial role for UPR in the establishment, progression, metastasis and chemoresistance of tumours as well as its involvement in the acquisition of other hallmarks of cancer. In this review, we will analyse the role of UPR in cancer development highlighting the ability of tumours to exploit UPR signalling to promote epithelial-mesenchymal transition (EMT).

## 1. Introduction

The ER is responsible for a large number of metabolic processes, including folding and post-translational modification of secretory proteins. Cells must keep a balance between the protein synthetic load and their capacity to ensure that folding and post-translational modifications are correctly performed. Improper ER function causes misfolding of de novo synthesized proteins and their accumulation due to a stringent quality control. Proteins failing to pass this control are returned to the cytosol and targeted for degradation by the ER-associated degradation system (ERAD) [1,2]. Under some physiological or pathological conditions, the capacity of the ER protein maturation machinery may be overwhelmed, leading to the accumulation of unfolded or misfolded proteins, an event referred to as ER stress.

The UPR is an adaptive mechanism evolved to relieve the ER stress restoring the metabolic and protein folding efficiency of the ER [3]. Activation of the UPR is initiated by the stimulation of three stress sensors that reside in the ER membrane: protein kinase R-like ER kinase (PERK), inositol requiring enzyme 1α (IRE1) and activating transcription factor 6 (ATF6) (Figure 1). Stress sensing is mainly dependent on GRP78 (HSPA5), an ER resident chaperone also known as BiP. Under homeostatic conditions, GRP78 is bound to the ER luminal part of the three sensors preventing their activation. Upon ER stress, the dissociation of GRP78 from the UPR sensors allows their activation initiating downstream signalling pathways that will help cells to cope with ER stress. Activation of the UPR reduces unfolded protein load through several pro-survival mechanisms, including the reduction of protein translation to decrease the ER overload, increase in the degradation of misfolded proteins via the ERAD system, and upregulated transcription of a large number of target genes to facilitate correct secretory protein maturation. However, if ER stress is not adequately solved and homeostasis is not restored, UPR may lead to a persistent signal that rather than reinstating ER homeostasis drives cells to apoptosis [2]. UPR is thus a complex mechanism that includes adaptive pro-survival and also pro-apoptotic responses. All three ER stress sensors trigger downstream signalling pathways that control survival or death decisions.

## 2. UPR Signalling Components

PERK is a type I transmembrane protein that, upon ER stress, dimerizes trans-autophosphorylates resulting in the activation of its kinase domain and the phosphorylation of eukaryotic translation initiation factor 2 alpha subunit (eIF2α) [4]. This phosphorylation transiently inhibits mRNA translation and attenuates global protein synthesis. This quick reduction of the ER overload relieves ER stress and has a pro-survival effect. Despite the overall reduction of protein synthesis, the phosphorylation of eIF2α allows the selective translation of a subset of UPR target proteins, including ATF4, a transcription factor that controls the expression of genes involved in protein folding, antioxidant response, amino acid metabolism and autophagy [5]. Active PERK also phosphorylates the nuclear factor erythroid related factor 2 (NRF2) which, upon translocation to the nucleus, controls the expression of anti-oxidant genes [6,7,8].

IRE1 is a type I transmembrane protein that contains two enzymatic activities on its cytosolic tail, a serine/threonine kinase and an endoribonuclease. In response to ER stress, IRE1 oligomerizes and trans-autophosphorylates undergoing a conformational change that activates its RNase domain. IRE1 RNase catalyses the unconventional splicing of an intron within XBP1 mRNA (XBP1u) shifting the XBP1 reading frame and producing a stable transcription factor known as XBP1s (thereafter XBP1) which promotes the transcription of genes involved in protein folding, ERAD, protein secretion and lipid synthesis [9,10,11]. Although IRE1 activity was first reported for its unconventional splicing activity, it is also involved in the degradation of ER-associated RNAs through a process known as RIDD (regulated IRE1-dependent decay) [12]. The molecular basis for the switch between RIDD and XBP1u mRNA splicing seems to be controlled by the oligomeric state of IRE1 [13,14,15]. Additionally, although less characterized, IRE1 activity also elicits the activation of the c-Jun N-terminal kinase (JNK) signalling [16].

ATF6 is a type II transmembrane protein that belongs to a family of transcription factors whose members contain a conserved bZIP domain on their cytosolic domain. Following ER stress ATF6 is transported from the ER to Golgi where it is cleaved by the proteases S1P and S2P [17]. The released ATF6 cytosolic domain (ATF6bZIP) translocates to the nucleus where it activates genes involved in ER quality control [18] and ERAD [19]. The cross-talk of ATF6 and XBP1 through heterodimerization further increases the scope of their target genes [2,20,21].

Several stresses both cell extrinsic and intrinsic may perturb the protein folding efficiency of the ER and lead to the accumulation of misfolded proteins, ER stress and UPR activation.

## 3. EMT: A Brief Update

EMT is a genetic and cellular programme that endows cells with mesenchymal features that ultimately facilitate motility (Figure 2) [22,23]. This reversible event was originally described while studying embryogenesis [24] and soon the EMT emerged as a crucial actor in tissue and organ development during morphogenesis and wound healing in adulthood [25]. EMT has since been linked to organ fibrosis and cancer, which is then referred to as pathological EMT [26]. In both physiological and pathological EMT, the expression of EMT transcription factors (EMT-TFs) launches a complex cellular programme that results in the loss of epithelial apical-basal cell polarity towards different degrees of mesenchymal morphology [27,28]. Essentially, the activation of EMT-TFs promotes a gene expression switch from genes involved in upholding epithelial cell polarity to genes responsible for the spindle-like morphology associated with mesenchymal features. These changes do not only relate to morphological traits, but also to the acquisition of several abilities that allow cells to move and invade nearby tissues [29].

It is nowadays well established that most carcinoma cells highjack the EMT programme to progress towards malignancy. In carcinomas, environmental cues and the exchange of signals among tumour cells and their microenvironment are mostly responsible for EMT implementation and the acquisition of mesenchymal traits and malignant progression. In fact, EMT is involved in the invasion-metastasis cascade, provides tumour-initiating abilities and contributes to the dormant state of disseminated tumour cells, cancer cell resistance to chemotherapy and immune evasion [22,23]. The core EMT-TFs responsible for implementing the EMT programme are the ZEB zinc finger TFs including ZEB1 and ZEB2, the zinc finger proteins belonging to the SNAIL family SNAI1 and SLUG (SNAI2), and the basic helix-loop-helix TFs TCF3, TWIST1 and TWIST2. These EMT-TFs can either bind directly to DNA or form transcriptional regulatory complexes to orchestrate the EMT by controlling the expression of numerous genes. ZEB, SNAIL, TCF3 and TWIST were originally described due to their ability to repress the invasion suppressor gene CDH1, encoding the cell adhesion protein E-cadherin [30], whose functional loss is considered a hallmark of EMT during carcinoma progression [31]. Hence, the transcription of EMT-TFs and their activity are tightly regulated by numerous layers and they act in distinct combinations to either activate or repress genes ultimately responsible for the features associated with a mesenchymal-like state during carcinoma progression [22,32]. Cell intrinsic and extrinsic signalling control the EMT programme by impinging on the expression and activity of the EMT-TFs and other cell-specific cofactors that act, in common or in non-redundant networks, to regulate the cellular plasticity associated with each particular context [22,29]. The reversion to the epithelial state through the mesenchymal-epithelial transition (MET) is responsible for metastatic outgrowth after tumour cell dissemination [33,34], characterized by the re-expression of epithelial markers and repression of mesenchymal traits [35]. Still poorly understood, MET regulation is most likely controlled by the shutdown of the aforementioned EMT-TFs, although signalling from the metastatic microenvironment may impinge on intracellular pathways to alter gene expression [23].

In cancer, increasing evidence indicates that the full implementation of the EMT programme, as defined in in vitro cellular models upon expression of an EMT-TF, would be a rare event. Besides, cells in different intermediate morphological states ranging from partially epithelial to quasi-mesenchymal are more likely to occur and be responsible for implementing different steps during tumour progression. Carcinoma cells undergoing EMT would thus show hybrid features and express a mixture of epithelial and mesenchymal markers, whereas incompletely losing their epithelial cell polarity. Indeed, these cells displaying partial EMT states have been documented in human cancers and are thought to be particularly plastic. These plastic EMT cells, while able to degrade and invade the surrounding stroma, may also, have given the signalling context, acquire stem cell-like traits, and become refractory to therapy and immune surveillance [22,27,36]. Moreover, these hybrid EMT cells are linked to increased metastatic potential in different mouse models of cancer, as well as in human tumours [27,37]. EMT was linked to breast cancer cell stemness [38,39] and its characteristic plasticity is now being related to cancer stem cell-like properties in different human tumours [27]. Cancer stem cells are also responsible for the appearance of tumour recurrence establishing a link between EMT and chemoresistance mechanisms, whereas tumour cells undergoing EMT have been shown to express immunosuppressive and immunoevasive molecules to avoid attack by the innate and adaptive immune systems in cancer mouse models [23].

Therefore, EMT endows subpopulations of cancer cells with a highly dynamic morphological plasticity tied to context-dependent functional abilities that facilitate malignant progression (Figure 2).

## 4. ER Stressors, UPR and EMT

Cancer cells are subjected to numerous intracellular and extracellular stresses that disturb ER homeostasis provoking ER stress and thus, UPR activation (Figure 1). During cancer progression the tumours acquire different biological properties including sustaining proliferative signalling, evading growth suppressors, resisting cell death, enabling replicative immortality, inducing angiogenesis, and activating invasion and metastasis. All of these biological properties constitute the so-called Hallmarks of Cancer [40]. In addition, to support the increased metabolic demand and the environmental pressure, cancer cells need to reprogramme their secretory functions to secrete metalloproteases, growth factors or cytokines that will facilitate tumour invasion and progression. Activation of the UPR influences this secretory switch [41]. Current evidence suggests that UPR activation favours tumour progression through the modulation of some of the Cancer Hallmarks [42]. One of the hallmarks powered by UPR signalling is the activation of the invasion-metastasis cascade, in which EMT plays a central role, as mentioned above [41,43,44,45,46,47,48]. Although a few reports suggest that EMT can in some instances result in UPR activation [49,50], the information is still scarce and this issue will not be addressed herein.

We will next describe the ER stressors that provoke UPR activation; linking how defined UPR signals modulate EMT (Figure 3).

### 4.1. Cell Extrinsic Stressors

#### 4.1.1. Chemicals

The UPR has been extensively studied in vitro by treating cells with chemical stressors such as thapsigargin, dithiothreitol or tunicamycin, which disturb calcium homeostasis, redox equilibrium or N-glycan synthesis, respectively. These drugs activate non-specifically the three arms of the UPR [51]. Also, ER stress can often be attributed to drug-induced adverse effects caused by numerous anti-cancer drugs used by present pharmacology such as bortezomib, cisplatin and doxorubicin, which impact the PERK and/or IRE1 branches of the UPR in different ways [52,53]. Recently, research efforts aimed at looking for IRE1 and PERK inhibitors as possible anti-tumoral drugs have intensified. In this sense, MKC8866 (IRE1 inhibitor) and ISRIB (p-eIF2α inhibitor), were shown to decrease breast cancer cell proliferation and promote tumour regression in patient-derived xenografts (PDX) from metastatic prostate cancer tumours, respectively [54,55].

EMT induction after UPR activation was first observed in vitro in rat alveolar epithelia cell lines upon treatment with the classical ER stressors tunicamycin and thapsigargin [56,57]. In both cases, activation of the IRE1 branch of the UPR leads to EMT in a SMAD2/3 and Src-dependent fashion although the underlying molecular mechanism was not analysed. These observations were further confirmed in vivo in a rat model of bleomycin-induced pulmonary fibrosis in which the analysis of bleomycin-treated animals revealed increased expression of IRE1-XBP1, mesenchymal cell markers, and the concomitant downregulation of epithelial cell markers [58]. These studies also showed that the promotion of EMT by XBP1 was dependent on SNAI1 expression in alveolar epithelia cell lines [58] (Figure 3).

The number of chemical stressors inducing EMT after UPR activation has been expanded by the recent finding that chemotherapeutic drugs activate ER stress and ultimately EMT [59]. Treatment of lung adenocarcinoma cell lines with cisplatin, cytarabine, doxorubicin, gemcitabine, vinorelbine or pemetrexed activate the PERK branch of the UPR which, in turn, induces EMT through the upregulation of SNAI1 and ZEB1 gene expression as well as EMT-like changes in several tissues in mice [59] (Figure 3).

#### 4.1.2. Hypoxia

Hypoxia compromises ER protein folding leading to the activation of the UPR [43] whereas it is also a known inducer of EMT in solid tumours [60].

Hypoxia activates the PERK arm of the UPR to promote metastasis in human cervical tumoral cells [61]. Hypoxic stress, through activated PERK, produces a rapid inhibition of protein translation due to the transient phosphorylation of eIF2α and the preferred translation of a subset of transcripts including ATF4 [5] (Figure 3). IRE1 has also been involved in facilitating cell survival under hypoxic conditions since XBP1-deficient cells exhibited reduced survival in vitro as compared with their wild-type counterparts when exposed to hypoxic environment [62]. Work in triple negative breast cancer (TNBC) cells [63] points to an interaction between XBP1 and hypoxia-inducing factor 1α (HIF1α) that form a transcriptional complex to promote efficient transcription of HIF1α target genes. By sustaining HIF1α transcriptional program, the IRE1-XBP1 arm of the UPR supports survival during hypoxia and sustains tumour growth [64] (Figure 3).

One consequence of the activation of the UPR in hypoxic tumours is an increase in autophagy. Through the liberation of amino-acids from long-lived proteins and the removal of damaged organelles, autophagy exerts a cytoprotective effect and helps cells to survive [43]. Both the PERK and IRE1 branches of the UPR participate in this survival mechanism. In several human cancer cell lines, hypoxia increased transcription of the essential autophagy genes microtubule-associated protein 1 light chain 3beta (LC3) and autophagy-related gene 5 (ATG5) through the activation of PERK [65]. Also, activation of IRE1 during the hypoxia-induced ER stress increases tumour cell tolerance to hypoxia [62] (Figure 3).

Activation of the UPR also plays a role in tumour adaptation to hypoxic stress by promoting angiogenesis. Angiogenesis is regulated by the secretion of soluble factors such as VEGF-A. Several examples implicate the IRE1-XBP1 axis in this process. In TNBC, XBP1 expression is required for HIF1α-mediated VEGF-A production and vessel formation under hypoxia, and xenografts derived from cells transfected with XBP1 shRNA displayed reduced angiogenesis [63]. Also, IRE1^-/-^ mouse embryo fibroblasts decreased the production of VEGF-A after ER stress [66]. In addition to IRE1, PERK signalling has also been involved in the upregulation of angiogenic factors. Xenografts derived from PERK KO cells showed delayed growth and the tumours presented reduced blood vessel formation [67]. More recently, it was shown that the PERK arm of the UPR controls the induction of angiogenic factors and that ATF4 itself binds to the promoter of VEGF-A [66,68].

The information about the axis hypoxia-UPR-EMT is still scarce, however several recent findings suggest that such a relationship exists. Gastric cancer cells under hypoxic conditions suffer EMT and the PERK and ATF6 arms of the UPR are activated. Moreover, knockout of PERK, ATF4 or ATF6 hinders the induction of EMT by hypoxia [69]. In TNBC, we have already mentioned that XBP1 drives tumorigenesis by assembling a transcriptional complex with HIF1α [63]. Perhaps the high metastatic potential of this subtype of breast cancer could be related to a positive action of HIF1α-XBP1 on EMT, whose occurrence in TNBC is well established [70]. Finally, the hypoxic activation of the PERK-eIF2α arm in human cervix cancer cells mentioned above was shown to upregulate LAMP3, a putative metastasis-promoting gene, which promotes both the migratory phenotype and the development of lymph node metastases suggesting the possibility of EMT occurrence [61].

#### 4.1.3. Nutrient Starvation

Intimately related with the insufficient vascularization of the tumour is the nutrient stress starvation suffered by tumour cells. The UPR participates in the rewiring of tumour metabolism by selectively activating catabolic pathways—we have already mentioned the induction of autophagy—but also biosynthetic pathways. In human tumour tissues, and several tumour cell lines, blocking UPR by silencing PERK or ATF4 significantly reduces the production of angiogenesis mediators induced by glucose deprivation [68]. Recently, oncogenic KRAS has been identified as a key regulator of the transcriptional response to nutrient deprivation in non-small cell lung cancer and ATF4, as a key transcription factor regulated by KRAS to support amino acid homeostasis. Through the regulation of ATF4, KRAS controls amino acid uptake and asparagine biosynthesis [71]. In addition to ATF4 signalling, IRE1 pathway activation of XBP1 controls the hexosamine biosynthetic pathway to generate substrates for protein glycosylation [72], and, in complex with HIF1α, activates a transcriptional programme to upregulate glycolytic enzymes and glucose transport [63].

Another recently published study has described that the availability of asparagine controls metastasis in breast cancer, at least in part through the modulation of EMT. Limiting asparagine by asparagine dietary restriction or by knocking down asparagine synthetase reduces the metastatic capacity of the primary tumour and, by contrast, an increase in dietary asparagine or enforced asparagine synthetase expression promotes metastatic progression. Interestingly, asparagine synthetase is a target of the PERK-eIF2α-ATF4 branch of the UPR and ATF4 knockdown mimics the phenotype caused by asparagine synthetase knockdown [73]. One plausible interpretation of these findings could be that asparagine starvation triggers activation of the PERK-eIF2α-ATF4 branch which in turn could upregulate asparagine synthetase gene expression and, thus, EMT (Figure 3).

Cancer cells and melanoma in particular require an exogenous supply of glutamine since it is commonly depleted in tumours. Interestingly, it has been recently demonstrated that glutamine starvation is responsible for an upregulation of ATF4 in melanoma cells [74]. In this study, the authors demonstrate that glutamine limitation promotes an ATF4 dependent repression of MITF, considered a melanocyte lineage differentiation gene. MITF downregulation in response to stress is then associated with enhanced invasion, upregulation of ZEB1 and downregulation of SNAI2, hallmarks of EMT-associated reprogramming in late-stage melanoma [75] (Figure 3).

Deprivation of essential amino acids also causes activation of the UPR but, in this case, the arms activated are the IRE1 and ATF6 [76]. In this study, the induction of EMT was not analysed, however, based on the mentioned role of XBP1 in upregulating SNAI1, starvation of essential amino acids could result in stress inducing the UPR and subsequently EMT. Glucose starvation also promotes EMT [77] but whether this EMT induction is a consequence of the expected UPR activation due to limited glucose availability is nowadays not known. Finally, a recent review discusses the evidences linking starvation-induced ER stress to invasiveness and cancer progression [78]. In tumours, besides nutrient limitation and hypoxia, other cell extrinsic or intrinsic cues would be able to trigger a translational reprogramming through eIF2α phosphorylation, which in turn would promote an invasive phenotype [78].

### 4.2. Cell Intrinsic Stressors

#### 4.2.1. Oncogenes

Malignant transformation by the activation of oncogenes is associated with an increased cell proliferation that imposes severe pressures on cellular processes such as an excessive demand for protein synthesis in the ER. When the folding capacity of the ER is exceeded, cancer cells trigger the UPR activating PERK, IRE1 and/or ATF6-dependent signalling pathways as shown in different types of cancer (Figure 1) [79,80,81,82]. The metabolic reset imposed by the oncogenic transformation establishes a complex interplay within the cancer cell among the different stress responses to promote cell survival and metastatic expansion. In some cases, UPR induction is clearly linked with a malignant phenotype and aggressiveness, as shown by XBP1 overexpression in human multiple myeloma [83], and overactivation of PERK-ATF4 in MYC-induced lymphomas [79], breast cancers [84] and colorectal adenocarcinomas [85]. Recent work with RAS-induced mouse primary tumours points to a complex role of IRE1 signalling. In mouse keratinocytes, RAS transformation promotes IRE1 activation and XBP1 splicing through the action of both ER stress and RAS-activated MEK-ERK signalling pathways, resulting in enhanced proliferation. However, following this initial proliferative response, keratinocytes stop growing and enter senescence prematurely. Senescence occurs in parallel with the dampening of the ER stress response and an increase in IRE1 supported by MEK-ERK signalling, indicating that in keratinocytes expressing oncogenic RAS, reduction of ER stress accelerates senescence dependent on IRE1-RIDD activity [82]. These results suggest that the type and the stage of the tumour may have an influence on the outcome of UPR activation either pro-survival or anti-tumorigenic.

In conclusion, the activation of the UPR upon oncogenic signalling is well documented but, in these cases, activation of the UPR pro-survival response is apparently unrelated to EMT induction.

#### 4.2.2. Oxidative Stress

Tumour cells accumulate increased levels of reactive oxygen species (ROS) that lead to the general phenomenon of oxidative stress. ROS are produced intracellularly by both enzymatic and non-enzymatic reactions in different cellular compartments during cell growth [86]. Cancer cells have much higher levels of ROS than normal cells [87] and, additionally, activation of PERK due to hypoxia results in upregulation of ER-oxidase ERO1α [88] to facilitate the oxidative protein folding in the ER. In fact, a correlation between the levels of ERO1α and poor prognosis in breast cancer patients has been reported [89]. Oxidative protein folding brings about, on the other hand, an increase in ROS production that may aggravate cellular stress. Therefore, activation of the UPR upon exposure to oxidative stress is an adaptive mechanism to preserve cell function and survival although persistent oxidative stress ultimately initiates apoptotic cascades [90].

This protective function of the UPR against oxidative stress has an important impact during the first steps of tumour invasion, when an abnormal activation of EMT causes tumour cell detachment and acquisition of an invasive and migratory phenotype. In this situation, PERK activation favours survival upon ECM detachment [91] and alleviates the oxidative stress concomitant to the loss of matrix attachment contributing also to metastasis in in vivo models [92] (Figure 3).

Cumulative data suggest that oxidative stress governs several phases of tumour progression [93] including EMT, as revealed by the finding that overexpression of NOX1, a NADPH oxidase that generates ROS, induces EMT [94]. Cancer cells upregulate multiple antioxidant systems to overcome the negative effect of oxidative stress [95]. As mentioned above, one of the defence mechanisms activated by oxidative stress is the PERK-eIF2α-ATF4 arm of the UPR [90,96], which in turn upregulates NRF2, a master regulator of the antioxidant response [97]. A direct link among oxidative stress, UPR and EMT is suggested by the recent finding that NRF2 promotes EMT in cancer cell lines [98,99] (Figure 3).

#### 4.2.3. Alterations in UPR Signalling Components

Different studies support a role for proteostasis imbalance in cancer development raising the possibility that alterations in the UPR itself may contribute to tumorigenesis [46,100]. Constitutive activation of UPR signalling components has been reported in various types of human cancers such as breast tumours, hepatocellular carcinomas or gastric tumours [101]. Overexpression of GRP78 is indicative of a more aggressive phenotype since it is overexpressed with higher frequency in high-grade estrogen-receptor-negative tumours than in low-grade estrogen-receptors-positive tumours [102] in line with the idea that UPR activation may play a protective role against apoptosis in tumour cells. In contrast, in lung adenocarcinoma a higher overall survival was shown for those patients displaying high expression of IRE1 [103].

Sequencing the genome of different types of human cancers exposed somatic mutations in IRE1. In fact, ERN1, the gene coding for IRE1, was found to rank fifth among all the genes coding for human protein kinases carrying driver mutations across various human cancers [104]. In a recent review [44], the authors describe the cancer-associated mutations identified in the three UPR stress sensors. Missense mutations are equally present in the three cases, silent mutations and in-frame deletions or insertions are enriched in IRE1 while nonsense mutations are more frequent in ATF6. Interestingly, the authors emphasize that not all somatic mutations are equally present in every cancer type but rather it appears to be some preferences, with IRE1 mutations being predominant in cancers from the nervous system, whereas gastrointestinal cancers are enriched in ATF6 and IRE1 mutations and urologic and lung cancers in ATF6 and PERK mutations. Unfortunately, the biological impact of these mutations on the expression, activity or stability of the ER sensors or, more importantly, on the tumour phenotype is still unknown in most cases. Significant advance has been reported in the case of some IRE1 mutations. In human cancer tissue samples of glioblastoma multiforme (GBM) two missense mutations (S769F and P336L) and one stop mutation (Q780stop) have been identified in IRE1 [104,105]. Very recently, the sequencing of additional GBM samples revealed a new somatic mutation in a less conserved amino acid (A414T) [106]. The authors analysed the differential impact of the IRE1 variants, expressed in a glioblastoma cell line, on their kinase and RNase activities and how they affected the cell phenotype, downstream signalling and gene expression profile. Together with the in vivo data of tumour development after expression of the IRE1 variants, the study demonstrates that mutations affecting IRE1 activity determine the development of GBM tumours [106]. While the IRE1-XBP1 splicing activity favours angiogenesis and higher expression of migration/invasion markers, IRE1-RIDD activity promotes attenuation of both responses in tumour cells, pointing to antagonistic roles of the two signalling outputs in GBM progression.

Nevertheless, although mutations in the distinct components of the UPR are present on different tumour types, knowledge on their role in cancer biology is still very limited.

#### 4.2.4. Protein Overexpression

Overexpression of individual proteins, not necessarily involved in signalling pathways driving tumour progression, can also provoke UPR activation and subsequent EMT induction.

One early report showed that enhanced expression of the protease inhibitor SERPINB3 promotes tumorigenesis and EMT via the PERK and ATF6 arms of the UPR which lead to NFκB activation and subsequent expression of the pro-tumorigenic cytokine IL6 [107].

More recently, it has been reported that the overexpression of LOXL2 also causes UPR activation and subsequent EMT induction [108]. LOXL2 is a secreted enzyme involved in covalent inter and intramolecular crosslinking of the extracellular matrix components but its involvement in intracellular processes and tumorigenesis has been increasingly postulated [109]. LOXL2 is retained within the ER when it is endogenously or ectopically overexpressed. In the ER, LOXL2 interacts with GRP78 which in turn activates the IRE1-XBP1 and PERK-eIF2α arms of the UPR [108]. Furthermore, the processed form of XBP1 is shown to bind to the promoters of *SNAI1*, *SNAI2*, *TCF3* and *ZEB2* activating their expression [108] (Figure 3). Interestingly, IRE1 inhibition hampers the upregulation of these EMT-TFs and blocks LOXL2 ability to induce a full EMT program [108]. Remarkably, in human tumours with overexpression of LOXL2, the protein is accumulated in structures that are compatible with an ER location and this subcellular localization pattern correlates with poor prognosis of squamous cell carcinomas and distant metastasis of basal breast carcinomas [110,111].

## 5. UPR and EMT Footprint in Human Tumours

The information reviewed above relative to the cooperation between the UPR and EMT during tumour progression comes mainly from data derived from experiments performed using cell lines and mouse models. Concerning clinical samples, thorough reviews have recently addressed the evidences of ER stress in human tumours by examining the expression levels of UPR signalling components [41,81]. As a matter of fact, UPR components have been detected in samples from brain, breast, colorectal, kidney, liver, lung, and pancreatic cancer patients and their overexpression has been mostly correlated with worse prognosis [41,81]. In B-cell hematological malignancies, the IRE1-XBP1 arm is essential due to the B-cell inherent secretory phenotype and GRP78 and/or XBP1 upregulation is associated with poorer outcome in leukaemia, lymphoma and multiple myeloma [81].

There are few studies to date that analyse in human cancer biopsies markers of EMT along with UPR activation markers. These studies would help to elucidate the prognostic value of both programmes in terms of patient outcome, therapy choice and/or treatment response. Recent works addressing both UPR and EMT pathways in clinical samples are summarized in Table 1. Some of these studies characterize the underlying molecular mechanisms in cellular models mostly supporting that UPR activation precedes EMT in tumour progression.

In this regard, the expression of an EMT signature is strongly correlated with ATF4 expression in datasets covering breast, colon, gastric, lung, and metastatic sites from patient tumour samples [49]. In colorectal carcinoma, there is an association of GRP78 and nuclear β-catenin staining at the invasive front of a small cohort of tumour tissues samples, suggestive of ER stress and EMT [50]. Also in colorectal carcinoma, higher IRE1 expression in patient samples is associated with lower overall survival and the molecular mechanism proposed is the activation of EMT by IRE1 [113]. Additionally, in hepatocellular carcinoma, the detection of XBP1 in tumour samples positively correlates with vimentin and negatively with E-cadherin [114]. In the case of glioblastoma, higher activity of the IRE1-XBP1 axis correlates with shorter patient survival, considerable tumour infiltration by immune cells and increased tumour angiogenesis and invasive properties. These tumour characteristics are associated with increased expression of EMT-related markers in primary derived glioblastoma cell lines [106]. In breast cancer, an EMT signature is the most relevant feature in tumours showing PERK activation [112]. The authors propose that the transcription factor CREB3L1, downstream of PERK, promotes metastasis particularly in those tumours showing activated PERK signalling and an EMT signature [112]. Lastly, in lung adenocarcinoma tumours, ER stress proteins such as IRE1 and PERK co-express with EMT markers in lung tumour samples compared to matched normal adjacent tissues [115].

A direct connection between ER stress and UPR activation with malignancy has been formally established in different cancer settings [41,46,47,81]. In many occasions, the authors did not directly address the role played by EMT in tumour progression even when UPR signalling was linked to increased invasion or other EMT-related roles. This was the case in TNBC [63], glioblastoma [116], pancreatic ductal adenocarcinoma [117] or esophageal squamous cell carcinoma [118]. The opposite scenario, analysis of UPR activation in tumours with EMT-related markers, is even less common despite the central role of ER stress signalling in cancer progression [43,81]. Moreover, the biological significance of EMT in tumour development has been long debated by pathologists partly due to the transient nature of EMT and the spectrum of EMT phenotypic states involved in different steps of the invasion-metastasis cascade, which limits the detection of EMT in clinical samples [23,27]. Additionally, data obtained from analyses such as next-generation sequencing or proteomic and transcriptomic profiling in tumour biopsies derive from heterogeneous tissues samples. This fact hinders the contribution of particular cell subpopulations bearing ER stress and different degrees of mesenchymal features. Besides, the key influence of tumour microenvironment signalling on sustaining UPR and EMT cannot be disregarded.

## 6. UPR and Therapeutic Opportunities

Cancer cells undergoing UPR activation and prone to EMT may sustain malignant progression since they are able to migrate and invade, display tumour-initiating properties and resilience in foreign microenvironments while they become skilled to evade drug therapy and immune surveillance. Nevertheless, the UPR can also be a pro-apoptotic signalling pathway. When cells are subjected to a chronic ER stress or cannot resolve the stress, the UPR directs the cells to apoptosis [2]. This fact constitutes a therapeutic opportunity in the treatment of cancer. In this sense, some new drugs provoking ER stress have shown their potential in pre-clinical models. An example is matrine that suppresses prostate cancer aggressiveness by inhibiting EMT and activating the UPR [119]. Acriflavine, an antibiotic with several anticancer effects, has been shown to interfere with EMT and ATF4 dependent UPR activation in pancreatic cancer cell lines restoring sensitivity to cytotoxic drugs [120]. In breast cancer cell lines, AECHL-1, a triterpernoid, has been suggested as an anti-neoplastic agent by inducing ER stress and suppressing EMT [121]. Similarly, another alkaloid, sinomenine, prevents glioblastoma cell proliferation and invasion by triggering ER stress and EMT suppression [122]. Finally, the use of novel iron chelators has been proposed as a promising anti-cancer therapy by regulating ER stress signalling and inhibiting the EMT programme through the metastasis suppressor protein NDRG1 [123].

## 7. Conclusions and Perspectives

The involvement of ER stress and the activation of the UPR in cancer initiation and progression is now supported both by in vitro analyses and data from clinical samples [41,43,46,81]. Whether the UPR implication in cancer is mediated through EMT is still debated, although ER stress and EMT are both linked to similar hallmarks responsible for tumour progression. In this review, we have highlighted the ER stressors that activate the UPR and subsequently EMT, suggesting that the UPR may be an additional upstream signal for the induction of the EMT programme; this role of the UPR might well be dependent on the tumour type and/or the nature of the triggering stress (Figure 3). Probably, there is an underestimation of the actual UPR-EMT axis contribution to cancer. The understanding of these allied pathways and the hierarchy governing their signalling as well as the stromal cues that support their activation during tumour progression is far from complete. In order to improve the clinical management of cancer patients, the identification of specific markers for UPR and EMT status in cancer cell subpopulations within tumour biopsies possibly will aid in predicting tumour progression and therapy response.

In conclusion, the development of drugs targeting ER stress in particularly susceptible cells such as those in different EMT states might constitute a promising cancer therapy.

## Figures and Tables

**Figure 1 jcm-08-00624-f001:**
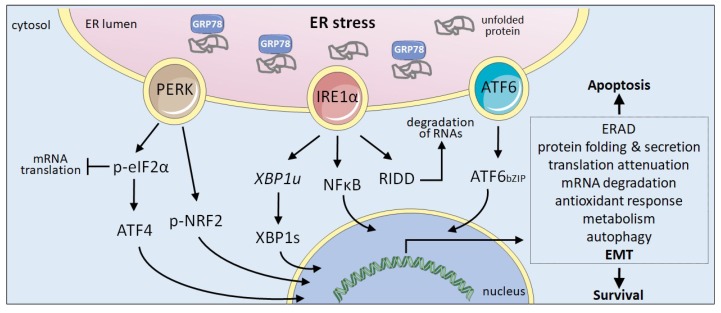
The UPR. The ER protein maturation capacity may be overwhelmed due to the action of several cell intrinsic and extrinsic factors, causing ER stress. The accumulation of unfolded proteins triggers the activation of the three ER-resident sensors responsible for UPR by sequestering GRP78. IRE1 mediates the unconventional splicing of the mRNA encoding XBP1 (XBP1u) rendering the functional transcription factor XBP1s and can activate NFκB signalling. IRE1 RNase degrades ER associated RNAs through RIDD (regulated IRE1-dependent decay). PERK phosphorylates eIF2α to inhibit global translation while promoting the translation of the transcription factor ATF4. PERK can also phosphorylate NRF2. ATF6 is exported from the ER to the Golgi apparatus, were the SP1 and SP2 proteases mediate the release of the bZIP domain (ATF6bZIP). In the nucleus, XBP1s, ATF4 and ATF6bZIP transcription factors trigger the expression of a large number of genes to help cells alleviate ER stress. Upon persistent ER stress, UPR favours apoptosis. Cancer cells exploit UPR signalling to promote survival under tumour-associated stress situations.

**Figure 2 jcm-08-00624-f002:**
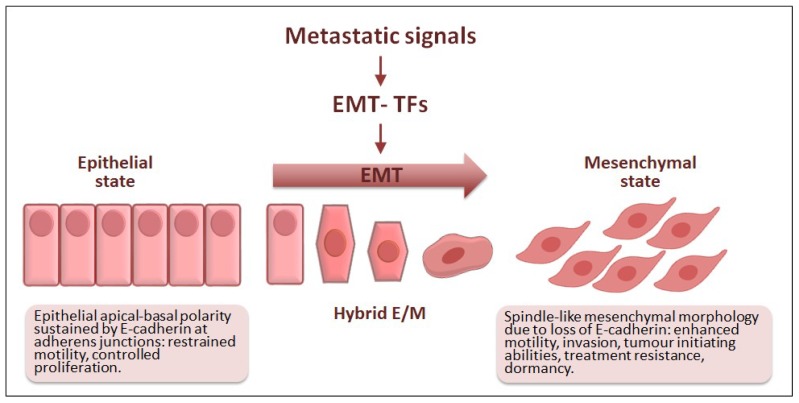
Epithelial-mesenchymal transition (EMT) in cancer progression. Different metastatic signals can activate one or more EMT-TFs which in turn trigger the EMT programme. During EMT, epithelial cells lose their apical-basal polarity and acquire mesenchymal traits that facilitate motility and contribute to the invasion-metastasis cascade. Some EMT-TFs directly control the expression of E-cadherin, whose functional loss is regarded as a hallmark of EMT. During EMT, hybrid epithelial/mesenchymal (E/M) states are also associated with tumour heterogeneity, tumour cell dissemination, cancer stem cell-like traits as well as immune evasion and resistance against conventional and targeted therapies.

**Figure 3 jcm-08-00624-f003:**
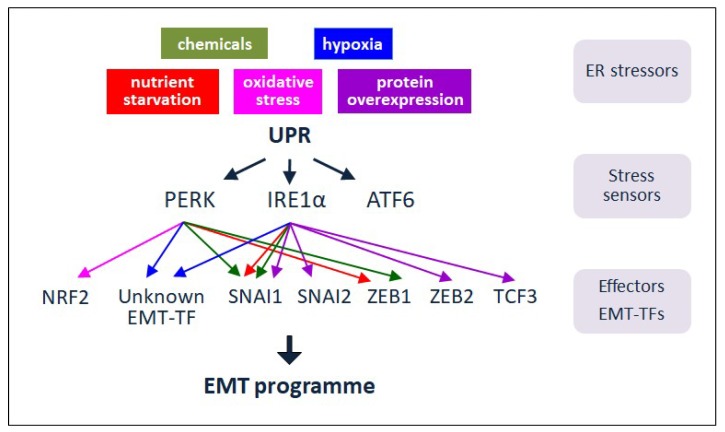
UPR signalling and EMT in cancer. In response to diverse ER stressors, UPR signalling is activated to relieve the stress and favour survival. In some tumours, the ER stress sensors and signalling players PERK and IRE1 are proposed to modulate EMT by impinging on particular EMT-TFs. Through the implementation of EMT, UPR can contribute to the progression and recurrence of tumours upon treatment. Thus, ER stress and UPR components can be exploited as plausible targets for anti-cancer therapy.

**Table 1 jcm-08-00624-t001:** Studies analysing UPR and EMT in clinical samples and/or primary derived cell lines.

Type of Cancer	UPR Activation	EMT Footprint	Source	Prognosis	Reference
Breast cancer	active PERK (ATF4 target genes)	EMT gene signature	human breast cancer datasets	NA	[49]
	active PERK (ATF4 target genes)	EMT gene signature	human breast cancer datasets	increased metastasis	[112]
Colon cancer	active PERK (ATF4 target genes)	EMT gene signature	human colon cancer datasets	NA	[49]
Colorectal carcinoma	IRE1	E-cad, N-cad	CRC tumour tissues and CRC cell lines	shorter overall survival	[113]
	GRP78	β-catenin	CRC tumour tissues	NA	[50]
Gastric cancer	active PERK (ATF4 target genes)	EMT gene signature	human gastric cancer datasets	NA	[49]
Glioblastoma	IRE1/XBP1 axis	VIM, ZEB1, TGFβ2	human GBM cancer datasets and primary derived GBM cell lines	shorter overall survival, increased tumour aggressiveness	[106]
Hepatocellular carcinoma	XBP1	VIM, E-cad	HCC tumour tissue	increased tumour size, increased metastasis	[114]
Lung cancer	active PERK (ATF4 target genes)	EMT gene signature	human cancer datasets	NA	[49]
	IRE1, PERK	ZEB1, SNAI2, SNAI1	LAC tumours	NA	[115]

CRC: colorectal carcinoma; GBM: glioblastoma; HCC: hepatocellular carcinoma; LAC: Lung adenocarcinoma; NA: not analysed.

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
