# Peer review of "UPR: An Upstream Signal to EMT Induction in Cancer"

_jcm, 2019, doi:10.3390/jcm8050624_

Reviewer 1 Report

The proposed manuscript "URP: An Upstream Signal to EMT Induction in Cancer” is a very interesting work. In this review the authors summarize the involvement of unfolded protein response in metastatic progression.

It is a good manuscript in which the issue is addressed in a clear and comprehensive way!

I have only one suggestion for authors: I think that the insert in the text of the images, as done in Figure 1, would help the reader to better focalize the biological pathways involved in EMT.

Author Response

Thank you very much for your suggestions that have greatly contributed to an increase in the quality of our manuscript. According to your comments we have included two new Figures (Figures 2 and 3)

Reviewer 2 Report

The paper “UPR: An Upstream Signal to EMT Induction in Cancer” by Santamaría PG et al is aiming to describe the role of the UPR in EMT induction. The paper is well written and easy to follow. The evidence linking the UPR to EMT is at this time weak, many times correlative, and controversial. The authors do acknowledge this but have raised some interesting ideas for further study.

I think the paper would benefit from additional graphics. Most of the paper focused on UPR and the mechanics of the UPR are illustrated nicely in figure 1. Less attention was focused on EMT and I feel graphic depictions of EMT including transcription factors and other important proteins, especially those discussed in the paper, might be very helpful to the reader. If possible a graphic demonstrating the links/relationship between the UPR (key proteins etc) and EMT would be helpful as well. 

Author Response

(The authors gave the same response as above.)
